# Adiabatic versus diabatic transport contributions to the ozone budget in the northern hemispheric upper troposphere and lower stratosphere

Frederik Harzer[1], Hella Garny[2], Felix Ploeger[3,4], J. Moritz Menken[2], and Thomas Birner[1,2]

[1]Meteorological Institute, Ludwig-Maximilians-Universität München, Munich, Germany
[2]Deutsches Zentrum für Luft- und Raumfahrt, Institut für Physik der Atmosphäre, Oberpfaffenhofen, Germany
[3]Institute of Climate and Energy Systems, Stratosphere (ICE-4), Forschungszentrum Jülich, Jülich, Germany
[4]Institute for Atmospheric and Environmental Research, University of Wuppertal, Wuppertal, Germany

**Correspondence:** Frederik Harzer (frederik.harzer@physik.lmu.de)

**Abstract.** Ozone in the extratropical lowermost stratosphere (LMS) is important for the local radiative balance and contributes to the tropospheric and near-surface ozone burden via stratosphere–troposphere exchange. Here, we aim to deepen our understanding of the transport contributions to LMS ozone in the Northern Hemisphere by studying the ozone budget in isentropic coordinates, which allows for a clean distinction of adiabatic and diabatic transport contributions. This is done by analyzing 20 years of ERA5 reanalysis output on model levels and a free-running simulation using the EMAC chemistry–climate model. Our analysis confirms that the ozone tendencies in the extratropical LMS at high latitudes are dominated by diabatic mean flow advection (associated with downwelling within the Brewer–Dobson circulation) and quasi-horizontal adiabatic eddy mixing due to planetary- and medium-scale Rossby waves. These transport contributions are somewhat weaker during summer compared to winter, although seasonality is found to be weaker in the LMS compared to higher altitudes. Horizontal mean flow advection is found to be relevant near the tropopause and just above the subtropical jet core. Notably, vertical (i. e., diabatic) eddy ozone transport is found to be important near the tropopause. While the adiabatic eddy ozone fluxes in the LMS are consistent with diffusive, down-gradient eddy transport, the vertical eddy ozone transport also features up-gradient regions, which by itself would act to reinforce the background ozone gradients near the tropopause. Closer analysis reveals that this is due to long-wave radiative damping of planetary waves, which acts to dampen the down-gradient horizontal eddy transport.

## 1 Introduction

Atmospheric ozone is known to impact life on Earth by its effects, e. g., on short-wave solar radiation, air quality and surface climate (e. g. WMO, 2022). It is mainly produced through photolysis in the tropical lower stratosphere (Chapman, 1929) and globally distributed throughout the stratosphere by the Brewer–Dobson circulation (BDC; e. g., Plumb, 2002; Butchart, 2014). Stratospheric ozone is not only important for the local radiative balance, but via stratosphere–troposphere exchange of air masses also contributes to the tropospheric ozone burden (e. g. Holton et al., 1995). This stratospheric contribution to tropospheric ozone crucially depends on the amount of ozone in the lowermost stratosphere (LMS; e. g., Albers et al., 2018). Since ozone in the LMS is primarily governed by transport, a detailed understanding of the different transport contributions is

critical and may ultimately also allow to consistently explain ozone trends derived from historical observations and from recent chemistry–climate simulations (e. g., Ball et al., 2019, 2020; Dietmüller et al., 2021; Benito-Barca et al., 2025).

Past research has shown that vertical advection by the diabatic mean flow (associated with up- and downwelling within the BDC), as well as adiabatic quasi-horizontal eddy mixing associated with dissipating Rossby waves are dominant transport contributions to lower stratospheric ozone (e. g., Plumb, 2002). At the same time, contributions by adiabatic horizontal mean flow advection and diabatic vertical eddy transport are usually considered to be less relevant. However, this picture may not be as valid near the extratropical tropopause due to the strong background gradients there (e. g., Gettelman et al., 2011) and the
potential impact of smaller scale tropospheric dynamics.

In this paper, we aim to improve our knowledge of the transport contributions to LMS ozone from a tracer budget perspective in isentropic coordinates. Here, the isentropic coordinate framework allows to cleanly separate diabatic and adiabatic transport contributions and may therefore help to isolate the effects associated with the individual transport processes in the upper troposphere and lower stratosphere (UTLS). To do so, we use recent ERA5 reanalysis data as a reference of the quasi-observed[1]
atmospheric circulation and historical climate variability. Since the representation of ozone may be problematic in ERA5 due to the simplified chemistry scheme and limited ozone data assimilation, we also study a free-running chemistry–climate model (CCM) with state-of-the-art representation of ozone chemistry. We document the climatologies of ozone transport in the LMS of the Northern Hemisphere, contrasting winter and summer, and work out structural characteristics of these transport contributions as a function of horizontal scale. Note that we do not intend to provide a thorough assessment of the performance
of the CCM compared to ERA5 in this study.

The structure of this paper is as follows: Section 2 describes the data and methods used in this work. In Sect. 3, we discuss the theoretical framework of the ozone budget approach and compare climatologies of the different ozone transport contributions. We then study aspects of eddy ozone transport separately and in more detail in Sect. 4. Section 5 provides the summary of our results and conclusions.

## 2  Data and methods

We use 6-hourly snapshots of ECMWF fifth generation reanalysis (ERA5) data on 137 vertical model levels with 1° horizontal resolution (Hersbach et al., 2017, 2020), which we interpolated onto isentropic levels with their vertical distance increasing with height (e. g., 39 levels between 300 K and 400 K with vertical resolution increasing from 1.25 K to 5 K). These isentropic levels were chosen to roughly match the vertical resolution of the underlying ECMWF Integrated Forecasting System (IFS)
based on a global mean climatology of potential temperature. We consider the time period 2000–2019 to align with the CCM simulation range (see below), which should be sufficiently long to study transport climatologies. We use updated ERA5.1 reanalysis data during 2000–2006, released by ECMWF due to substantial temperature biases in the lower stratosphere found in ERA5 for that time (Simmons et al., 2020). ERA5 is based on ECMWF's IFS version Cy41r2, which uses the linearized ozone parameterization by Cariolle and Teyssèdre (2007) and includes different monthly ozone climatologies based on ex-

---

[1] i. e., the reanalysis output is broadly constrained by observational records through data assimilation

ternal reanalysis data to be considered in the radiation parametrization (ECMWF, 2016; Hersbach et al., 2020). Davis et al. (2017, 2022) showed that stratospheric ozone in ERA5 agrees reasonably well with observations, especially in more recent years since 2004 when Aura MLS satellite measurements (Waters et al., 2006) were available for data assimilation. While we do expect increased uncertainties in the ERA5 output prior to 2004 due to the missing MLS records, a first estimate on sensitivities provided evidence that this does not have considerable effects on the results discussed in this paper (not shown).

Note that some variables in ERA5 are only provided as IFS model forecasts, initialized with the analysis twice daily at 06 and 18 UTC, and are not subject to direct assimilation of observational data. In particular, this holds true for the total diabatic heating rates examined in this study: these were derived from the ERA5 forecast data (variable "time-mean temperature tendency due to parametrisations", labeled "mttpm" in ERA5; cf. ECMWF, 2025), and include all contributions associated with long-wave and short-wave radiation under full-sky conditions, latent heat release, and heating due to turbulence and mixing processes. These

model forecasts run over 12 hours and output is available every hour. The forecast temperature tendency was converted into a diabatic heating rate following Ploeger et al. (2021), including an averaging over 6-hour windows centered on each synoptic time (00, 06, 12, and 18 UTC) to accurately represent 6-hourly mean heating rates. For further process analysis, diabatic heating rates have been derived similarly for full-sky and clear-sky radiative heating only, respectively.

  Furthermore, we consider 5-hourly instantaneous global model output from a free-running ECHAM/MESSy Atmospheric

Chemistry (EMAC) simulation (Roeckner et al., 2003, 2006; Jöckel et al., 2010) for the time period 2000–2019. This simulation is initialized in 1998 for a two-year spin-up phase. It is forced by sea surface temperatures and sea ice concentrations taken from ERA5 reanalysis data and was performed with a spherical truncation T42 of the spectral dynamical core, corresponding to a quadratic Gaussian grid horizontal resolution of approx. 2.8° in latitude and longitude, and with 90 vertical hybrid sigma-pressure levels covering the atmosphere from the surface up to the mesosphere. The overall model configuration originates from

a previous EMAC simulation prepared for the Chemistry–Climate Model Initiative (CCMI-2022), which itself was based on the setup used within the "Earth System Chemistry integrated Modelling" (ESCiMo) initiative (Jöckel et al., 2016) and updated to align with the new CCMI-2022 guidelines and recent submodel developments (Jöckel et al., 2024). Major differences between that setup for CCMI-2022 and the EMAC configuration used here, next to further submodel updates, refer to the additional application of the tropospheric aerosol model GMXe (Pringle et al., 2010) and the boundary conditions for ozone-depleting

substances. The latter differ from the actual CCMI-2022 requirements (e. g., Plummer et al., 2021): until 2014, they are taken only from observation-based Climate Model Intercomparison Project phase 6 (CMIP6) forcing data (Meinshausen et al., 2017) and include the species $CH_2Cl_2$ and $CHCl_3$. From 2015 onwards, we apply the boundary conditions for greenhouse gases and ozone-depleting substances from the CMIP6 SSP2-4.5 scenario (Meinshausen et al., 2020).

  A complete list of the EMAC submodels used in our model configuration is provided in the supplemental material of

this paper (Table S1). The simulation features interactive chemistry and online computation of the radiation budget based on instantaneous tracer field values (Sander et al., 2014, 2019; Dietmüller et al., 2016). Note, however, that the total diabatic heating rates were not available in the model output and therefore had to be approximated from 5-hourly model data afterwards by computing the total derivative of the temperature field on model levels for each time step. Furthermore, although EMAC's vertical resolution is slightly lower than that of ERA5, we interpolated the EMAC model output onto the same isentropic levels

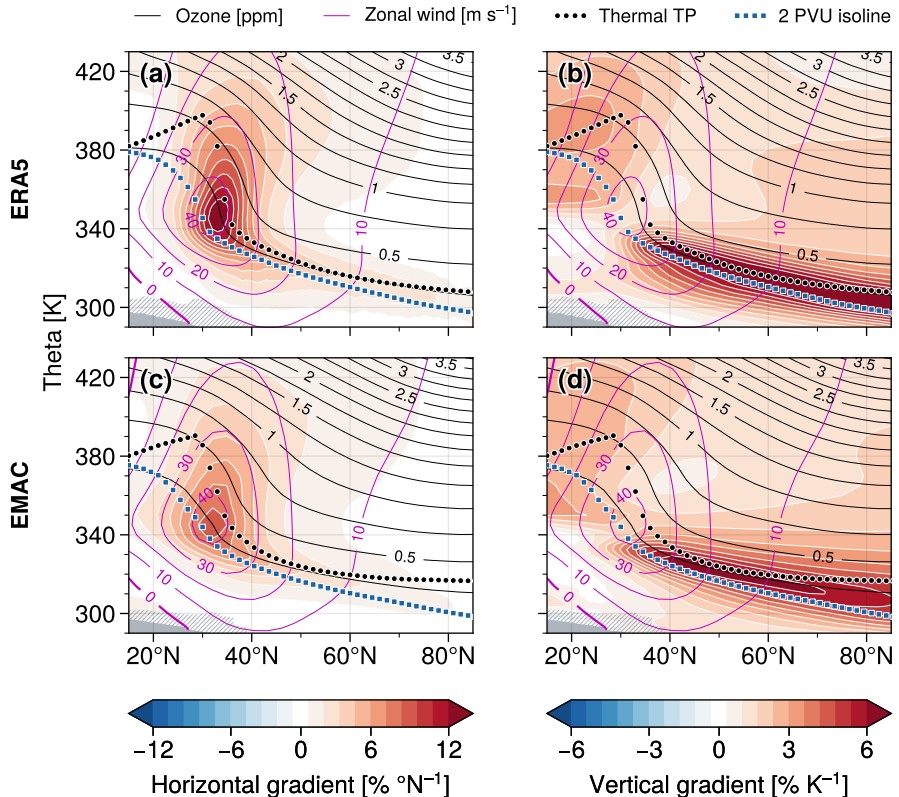

**Figure 1.** Winter-mean climatologies of (**a**) horizontal and (**b**) vertical zonal-mean ozone gradients (colors), respectively, as obtained from ERA5 reanalysis data (DJF, 2000–2019). (**c, d**) Same as in (**a, b**) but for the EMAC simulation. Gradient magnitudes are provided relative to the corresponding winter-mean, density-weighted ozone climatology (shown by the black contours). Violet contour lines represent climatologies of the zonal-mean zonal wind. Thermal tropopause heights following the WMO (1957) tropopause definition and 2 PVU potential vorticity isolines are shown by the thick dotted, black and blue meridional profiles, respectively. Grey hatches close to the surface indicate grid points in the lower troposphere where monthly-mean data were missing for more than 40 % of the winter months considered here.

as for ERA5, in order to retain ERA5's full resolution and at the same time facilitate the comparison of the individual analyses. We did not find any significant differences due to this slight vertical oversampling of the EMAC data (note that the vertical distribution of the EMAC model levels is still quite dense across the LMS).

Figure 1 provides an overview of the winter-mean ozone gradients in the northern hemispheric UTLS relative to climatological ozone for ERA5 and EMAC (DJF, 2000–2019). The relevance of these gradients for ozone transport through the
residual stratospheric circulation and eddy dynamics will be discussed in detail in subsequent sections. At this point, we note that the local maxima of both the horizontal and vertical gradients in the subtropics and along the extratropical tropopause are substantially weaker in EMAC compared to ERA5. This is likely due to the much coarser resolution in EMAC compared to in ERA5.

## 3 Climatologies of the transport contributions to the LMS ozone budget

The general expression of local changes of a given zonal-mean tracer distribution $\overline{\chi}^*$ with time can be derived by combining the mass conservation and tracer continuity equations, and in isentropic spherical coordinates with geographical latitude $\phi$ and potential temperature $\theta$ as the horizontal and vertical coordinates, respectively, reads (Andrews et al., 1987; Plumb, 2002)

$$\partial_t \overline{\chi}^* + \frac{\overline{v}^*}{a} \partial_\phi \overline{\chi}^* + \overline{Q}^* \partial_\theta \overline{\chi}^* = -\overline{\rho_\theta}^{-1} \left[ \frac{1}{a\cos\phi} \partial_\phi \left( \overline{\hat{v}\rho_\theta\hat{\chi}} \cos\phi \right) + \partial_\theta \overline{\hat{Q}\rho_\theta\hat{\chi}} \right] + \overline{S}^*. \tag{1}$$

Here, $\rho_\theta = -g^{-1}\partial_\theta p$ is isentropic density, overbars with asterisks indicate density-weighted zonal averages, e. g. $\overline{\chi}^* = \overline{\rho_\theta \chi}/\overline{\rho_\theta}$,

and hat symbols denote the deviations from these density-weighted, zonal-mean quantities. The overbars without asterisks refer to the conventional zonal mean along latitude circles. Furthermore, $a$ in Eq. (1) denotes the earth's radius, $v$ is the meridional wind (i. e., the adiabatic component in isentropic coordinates) and $Q = d\theta/dt$ indicates diabatic heating (i. e., vertical motion in isentropic coordinates associated with changes of potential temperature). With that, the second and third term on the left-hand side of this equation represent advective transport by the horizontal (adiabatic) and vertical (diabatic) mean flow, respectively. The first and second term on the right-hand side correspond to isentropic (i. e., adiabatic) and diabatic

eddy transport, respectively. Finally, $\overline{S}^*$ accounts for the effects of chemical sources and sinks of the tracer.

### 3.1 Winter-mean ozone transport

Figure 2 provides the northern hemisphere winter-mean ozone tendencies in the UTLS associated with the different transport contributions according to Eq. (1), contrasting results for ERA5 with EMAC. Overall, we find good agreement between the

ERA5 and EMAC climatologies for each contribution, lending confidence to the robustness of the results. The ozone tendencies associated with vertical mean flow advection in the first row of Fig. 2 support our understanding of the stratospheric residual circulation, which transports ozone-rich air from higher altitudes into the extratropical LMS, with a tendency to accumulate ozone there throughout the winter season. These tendencies are somewhat inhomogeneous in the extratropical tropopause region in EMAC compared to ERA5 (cf. Fig. 2e vs. 2a), which may in part be due to uncertainties in off-line derived diabatic

heating rates in EMAC. Furthermore, we note that these tendencies are substantially weaker at the tropopause in EMAC than in ERA5, consistent with the reduced vertical gradient magnitudes in EMAC (cf. Fig. 1b,d). We find a similar effect for the quasi-horizontal eddy transport contributions in the second row of Fig. 2, which feature a clear dipole signature centered around the extratropical tropopause with positive ozone tendencies in the subtropical upper troposphere and negative tendencies in the mid-latitude and polar LMS. These are consistent with horizontal two-way eddy mixing of ozone across positive meridional

background gradients. Again, the peak tendencies near the tropopause appear to be reduced in EMAC compared to ERA5, whereas we find similar results in the lower stratosphere for altitudes higher than $\sim 400\,\mathrm{K}$.

Overall, the ozone tendencies associated with vertical transport by the mean flow and with horizontal eddy mixing illustrate that these are the most important mechanisms of stratospheric ozone transport during northern hemispheric winters. The third and fourth row of Fig. 2 suggest that the other transport contributions are relevant mostly in the subtropical UTLS and near

the extratropical tropopause: first, panels 2c and 2g show negative tendencies due to meridional mean flow advection along the

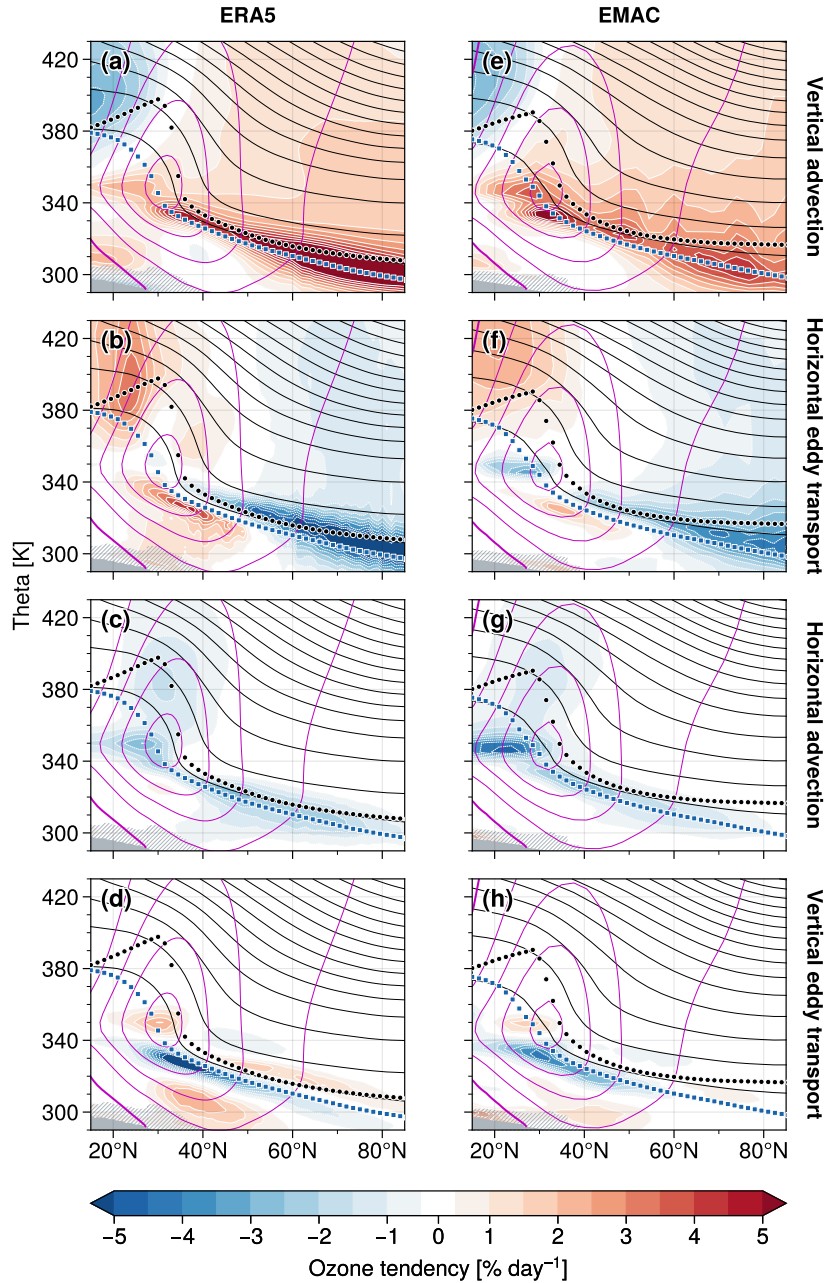

**Figure 2.** Ozone tendencies (colors) associated with mean flow advection and eddy transport in (**a–d**) ERA5 and (**e–h**) EMAC (DJF, 2000–2019), following the ozone budget approach from Eq. (1). Tendencies are given in units of % day$^{-1}$ relative to the corresponding zonal-mean ozone climatology. Other details as in Fig. 1.

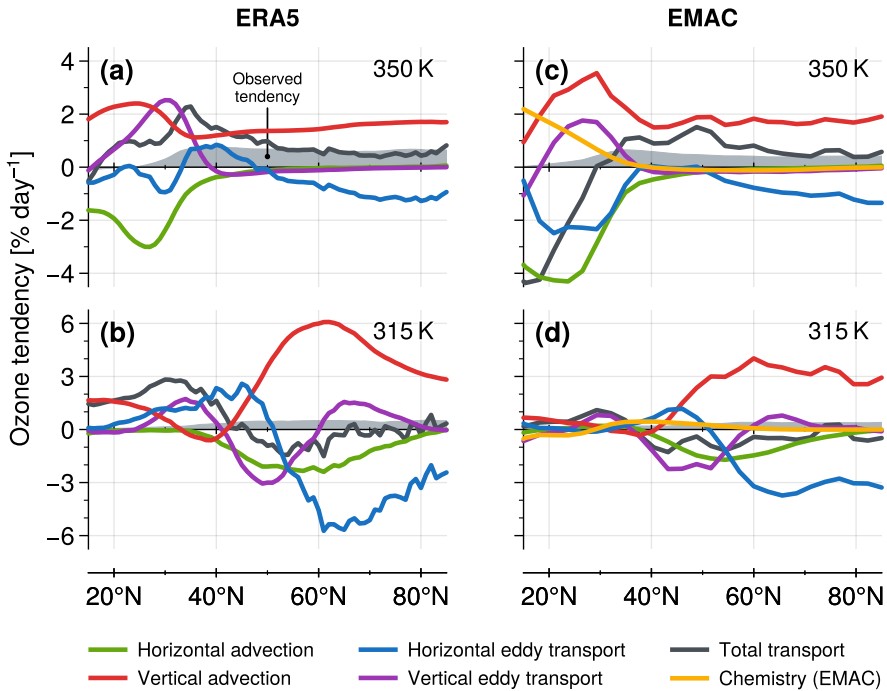

**Figure 3.** Meridional profiles of the ozone tendencies shown in Fig. 2 at 350 K (top row) and 315 K (bottom row). The net transport tendencies, accounting for all four transport contribution terms in Eq. (1), are shown as black lines. The tendencies due to ozone chemistry in EMAC are plotted yellow. Grey shading shows the observed winter-mean tendency derived from the actual ozone distribution.

tropopause and above the core of the subtropical jet stream (STJ). This is consistent with the upper branch of the tropospheric residual circulation acting on positive horizontal gradients of the background ozone distribution, i. e., the poleward mean flow is shifting air masses with smaller ozone mixing ratios towards higher latitudes. Second, vertical eddy transport causes a complex pattern of several local tendency maxima and minima along the tropopause and in the upper troposphere (Fig. 2d,h). At this point, we refer to the more thorough discussion of eddy ozone transport in Sect. 4 below.

It is interesting to note the stronger negative ozone tendencies equatorward of the STJ core around 350 K in EMAC compared to ERA5, both for meridional mean flow advection (panel 2g vs. 2c) and horizontal eddy transport (2f vs. 2b). These may be associated with the somewhat stronger wave propagation from the equator towards subtropical latitudes in the upper troposphere in EMAC (not shown), which may hint at increased equatorial Rossby wave activity. The increased horizontal ozone transport is likely also supported by the slightly larger horizontal ozone gradients at the tropical upper troposphere in EMAC compared to ERA5 (cf. Fig. 1a vs. 1c around 350 K and equatorward of ∼30°N), possibly linked to biases in the simulation of tropospheric ozone (cf. Jöckel et al., 2016).

For a more quantitative comparison of the different ozone budget contributions, Fig. 3 provides meridional profiles of the climatological tendencies on two selected isentropic levels in the UTLS: 350 K, which connects the upper subtropical troposphere

and lower extratropical stratosphere across the STJ core, as well as 315 K, which crosses the troposphere in the subtropics and the tropopause region in higher latitudes. Figure 3 also allows to compare the sum of the individual transport contributions (shown by the black curves) to the actual, "observed" winter-mean ozone tendency (indicated by the grey shading) derived from the underlying zonal-mean ozone distribution. For EMAC, we furthermore consider the available tendencies associated with ozone chemistry, $\overline{S}^*$ in Eq. (1), shown by the yellow curves. Figure 3 shows that the observed ozone tendency (grey shading) at 350 K in the LMS is mainly reproduced by the combined effects of vertical (diabatic) mean-flow advection and horizontal (adiabatic) eddy transport at higher latitudes (i. e., north of ∼60°N). In contrast, for tropical to subtropical latitudes at 350 K, additional contributions by horizontal advection, vertical eddy transport and ozone chemistry become relevant. Here, the differences between ERA5 and EMAC in the subtropical upper troposphere suggest that reproducing the substantial contributions by tropospheric ozone chemistry and the complex interactions between the individual transport processes is rather challenging in that region. At 315 K, horizontal mean-flow advection and vertical eddy ozone transport contribute substantially even at extratropical latitudes, while the effects by ozone chemistry are rather small. Here, the meridional structures of the different transport contributions are qualitatively similar for ERA5 and EMAC, whereas their local magnitudes are much larger for ERA5, again likely due to the stronger background ozone gradients along the tropopause there compared to EMAC.

Overall, the mostly close agreement between the sum of all transport contributions (black curves in Fig. 3) and the actual, observed ozone tendency (grey shading) northward of approx. 60°N and especially at 350 K suggests the approximate closure of the local ozone budget in the polar LMS [cf. Eq. (1)]. Note that this is not necessarily expected, e. g., due to off-line computation of the underlying equations, sensitivity to parameterizations in both models, or additional data assimilation in the reanalysis approach. The substantial differences between ERA5 and EMAC in the tropics at 350 K may be due to different representations of large-scale tropical dynamics as well as due to differences in the vertical location of main convective outflow (e. g., Fueglistaler et al., 2009).

Figure S1 in the supplement provides the ozone tendency profiles at 400 K and 800 K, illustrating the transport contributions by the shallow and deep branch of the residual BDC in the lower and middle stratosphere, respectively (e. g., Plumb, 2002; Birner and Bönisch, 2011; Baikhadzhaev et al., 2025). We find that substantial horizontal eddy transport in middle and higher latitudes is balanced mainly by vertical mean-flow advection at 400 K and ozone chemistry at 800 K, respectively, resulting in only very small total ozone tendencies during Boreal winter there. Furthermore, it is worth noting that meridional mean-flow advection removes (supplies) substantial amounts of ozone in the subtropics at 400 K (800 K), which is of potential relevance for supporting horizontal, wave-driven ozone transport by the BDC.

### 3.2 Seasonal variations of stratospheric ozone transport

Transport in the stratosphere is subject to substantial seasonal variability. In the following, we illustrate some aspects of seasonal variations of the different transport contributions to the zonal-mean ozone budget in the LMS.

Figure 4 shows the climatological ozone tendencies associated with advective transport and eddy dynamics as in Fig. 2, but now for northern hemispheric summer (June–August, 2000–2019). We find substantially reduced zonal-mean ozone gradients in the LMS compared to the winter months (black contours in Fig. 4) and reduced zonal wind speeds in the subtropical jet

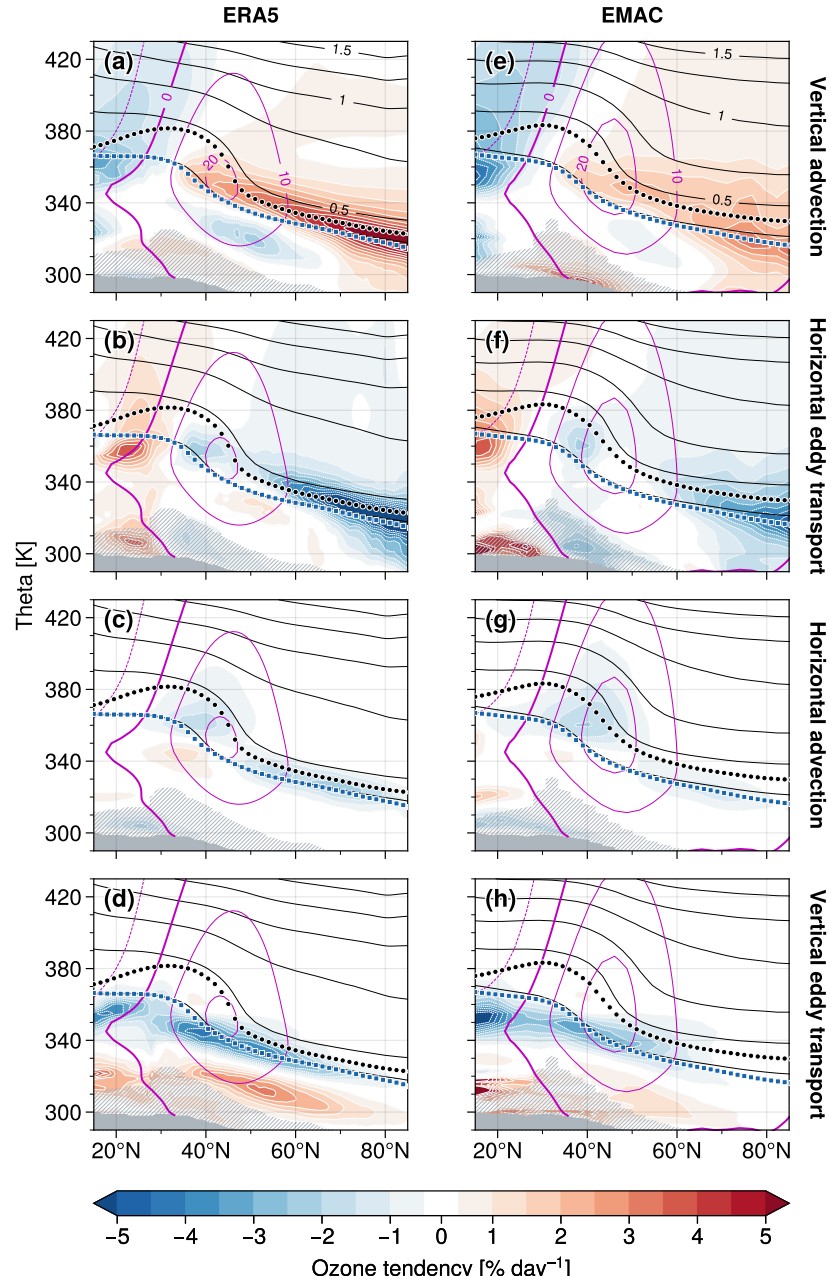

**Figure 4.** Same as Fig. 2 but for northern hemispheric summer (June-July-August).

stream region that come along with a weaker tropopause break. The overall structures of the ozone tendencies, however, are qualitatively quite similar to those of Boreal winter (cf. Fig. 2). We note the following details in the climatologies for the summer season:

– substantial contributions by vertical advection due to the mean flow and horizontal eddy mixing in the tropopause region (of similar magnitude compared to winter, compare first and second rows of Fig. 2 and Fig. 4) as part of the shallow branch of the BDC, despite the expected strongly reduced tendencies at higher altitudes (above $\sim 360$ K)

– strong dipole tendencies due to horizontal eddy mixing between the tropical upper troposphere and subtropical tropopause equatorward of $\sim 40°$N (second row), likely reflecting local Monsoon effects in the zonal-mean perspective (cf. Konopka et al., 2010; Abalos et al., 2013)

– reduced negative ozone tendencies due to meridional mean flow advection (third row) compared to winter, which is consistent with both a weaker residual tropospheric circulation and weaker horizontal ozone gradients

– strong dipole signature in tropospheric and near-tropopause ozone tendencies due to vertical eddy transport (fourth row), indicating downward transport of stratospheric air masses with higher ozone concentrations that is even stronger compared to winter (cf. Yang et al., 2016)

Next, in Fig. 5 we study the climatological meridional structure of the ozone transport contributions associated with mean flow advection and horizontal eddy transport as a function of time during the year at the isentrope that is located 50 K above the STJ core ($\approx 400$ K during Boreal winter), which is intended to follow the seasonality of the shallow branch of the BDC in the lower stratosphere on a monthly basis (see the figure caption for more details on the computation). The results for the 800 K isentrope, tracking the seasonal cycle of the deep branch of the BDC, are available in the supplemental material. We do not consider vertical eddy mixing here since the associated contributions are rather small at these levels (cf. Fig. 3).

In the upper two rows of Fig. 5, both ERA5 and EMAC show substantial equatorward eddy ozone transport across the subtropics throughout the year, with somewhat stronger horizontal transport during boreal winter and spring. The transport contributions by medium- to smaller-scale waves (wave numbers 4+) turn out to be larger than those associated with planetary-scale waves (wave numbers 1–3). Furthermore, the dipole transport pattern for wave numbers 4+ follows the latitudinal shifting of the STJ throughout the year (green curves in Fig. 5), which is different to the transport signatures associated with the planetary-scale waves. For the latter, we instead find slightly negative tendencies around 40°N during early summer, likely reflecting local Monsoon effects (cf. second row in Fig. 4). This is consistent with the high zonal wind speeds in the STJ region representing an effective barrier for tracer transport, as also illustrated by the time-dependent location of the meridional gradient maxima of potential vorticity (blue dashed lines in Fig. 5) that are closely aligned with the STJ core.

Panels 5c,g show negative ozone tendencies in the subtropics due to horizontal mean flow advection, which are somewhat weaker during summer. The seasonal cycle is in close correspondence with that of medium- to smaller-scale horizontal eddy transport (second row of Fig. 5), since both the residual flow and horizontal mixing are driven by wave dissipation at mid-latitudes (e. g., Plumb, 2002). We found that the meridional mean flow in the lower stratosphere is slightly weaker during the summer months but continuously directed polewards throughout the whole year (not shown), suggesting rather minor seasonal changes in the dynamical drivers of transport there. Instead, the seasonality in ozone transport due to meridional advection and horizontal mixing seem to be mainly associated with seasonal variations of the strength of the STJ and, as a consequence,

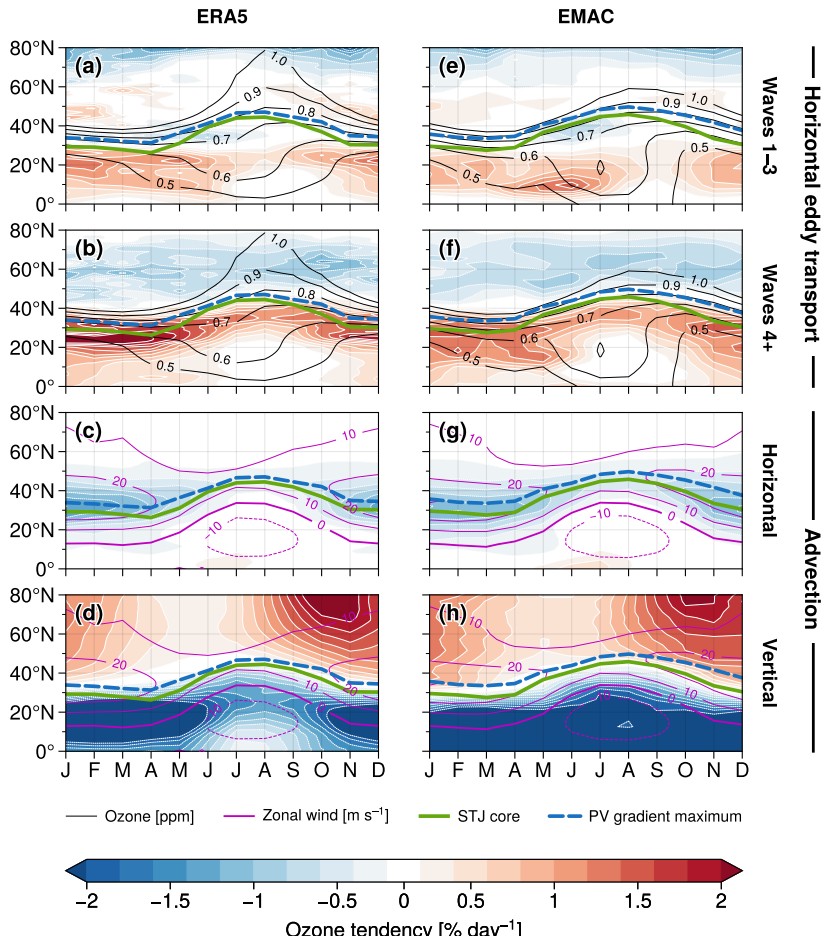

**Figure 5.** Climatologies of ozone tendencies (colors) associated with advection and horizontal eddy transport as a function of month during the year, each interpolated onto the isentrope 50 K above the STJ core, for ERA5 (left column) and EMAC (right column). Black contour lines in the upper two rows show the seasonal evolution of subtropical ozone (contours displayed for ozone values between 0.5 ppm and 1.0 ppm only). The violet contours provide the zonal-mean zonal wind climatologies. The green thick curves illustrate the meridional location of the STJ core, which corresponds to the maximum zonal wind shear relative to the 850 hPa zonal wind following the method by Davis and Birner (2013, 2017). The height of the STJ core is estimated from the corresponding zero-crossing of the vertical derivative (in log-$p$ coordinates) of the zonal wind profile interpolated at the STJ's meridional position. The blue dashed lines show the position of the maximum meridional gradient of potential vorticity in the subtropics (derived from the corresponding zero-crossing of its meridional derivative) at the respective isentrope 50 K above the STJ.

215  subtropical ozone gradients that limit the effectiveness of dynamical transport (ozone climatology shown by black contours in Fig. 5). In contrast, the substantial seasonal cycle of diabatic mean flow advection (panels 5d,h) likely is governed by the

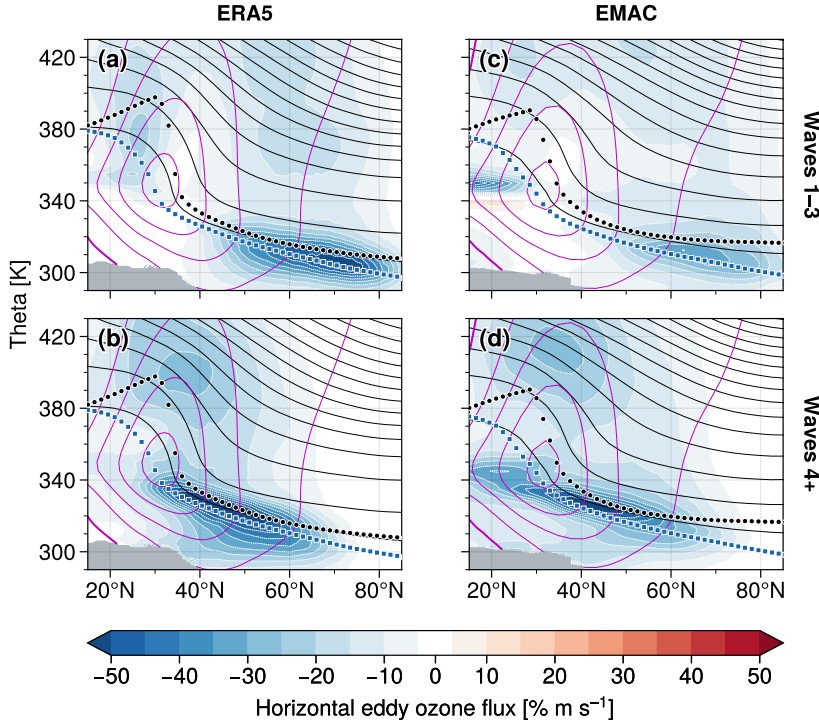

**Figure 6.** Horizontal eddy ozone flux (colors) associated with planetary-scale waves (top row, zonal wave numbers 1–3) and medium- to smaller-scale waves (bottom row, wave numbers 4+), as derived from ERA5 reanalyses (left column) and EMAC data (right column), respectively (winter-mean climatologies, DJF 2000–2019). Other details as in Fig. 1. A more detailed wave decomposition is provided in Fig. S3 in the supplemental material.

strong seasonality of large-scale planetary waves that propagate into the higher stratosphere (cf. Fig. S2 in the supplement) and that drive the deep branch of the BDC primarily during winter (Charney and Drazin, 1961; Plumb, 2002).

## 4  Eddy ozone transport in the UTLS

220 In this section, we have a closer look at the eddy contributions to the zonal-mean ozone budget in the LMS. Figure 6 provides the individual winter-mean *horizontal* eddy ozone fluxes associated with planetary waves (zonal wave numbers 1–3) and medium- to smaller-scale waves (wave numbers 4+)[2]. Both ERA5 and EMAC show overall similar patterns with negative meridional ozone fluxes almost everywhere in the LMS, indicating equatorward eddy ozone transport (cf. Figs. 2 and 5), which is consistent with the well-known picture of diffusive, down-gradient eddy transport along the positive horizontal gradients of

225 the background ozone distribution. We find the strongest ozone fluxes due to planetary-scale waves in the mid- to high latitudes along the tropopause, whereas medium- and smaller-scale wave activity acts mainly in the subtropics to mid-latitudes. However,

---

[2]derived from Fast Fourier Transformations (Virtanen et al., 2020) along latitude circles; cf. supplemental material for more detailed wave decompositions

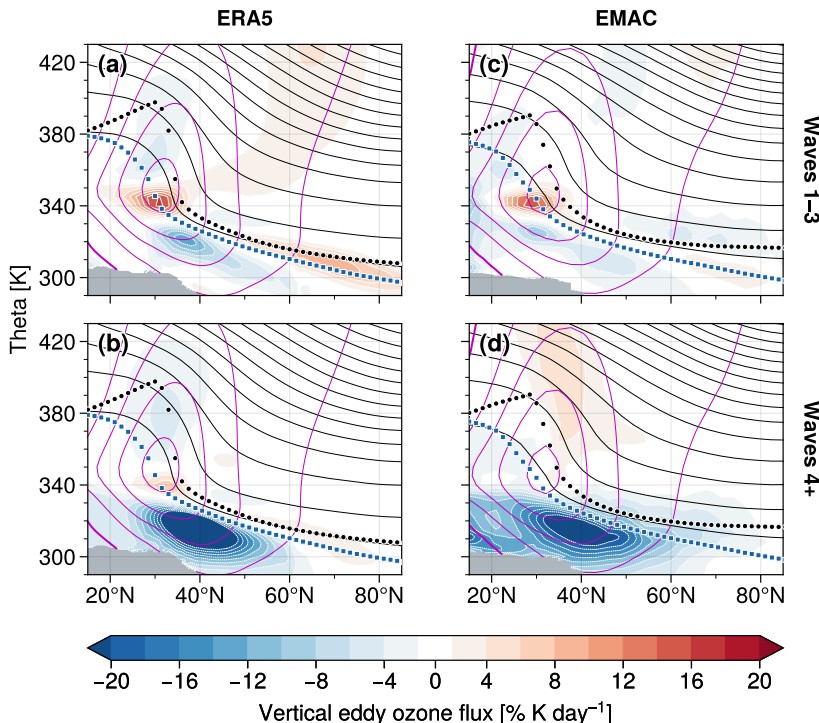

**Figure 7.** Same as in Fig. 6 but for the vertical eddy ozone flux. See Fig. S4 in the supplement for a more detailed wave decomposition.

since geometric wave lengths for a given zonal wave number decrease toward higher latitudes, we interpret the substantial negative ozone fluxes due to wave numbers 1–3 in the polar LMS also to be due to medium- and synoptic-scale wave activity.

For Boreal summer (June–August), we found somewhat weaker but still substantial horizontal eddy ozone transport in the tropopause region compared to winter (cf. Fig. S5). In the lower stratosphere, medium- to smaller scale waves contribute almost equally as during winter, whereas horizontal ozone fluxes associated with planetary waves are substantially reduced.

Figure 7 provides the winter-mean wave decompositions of the *vertical* eddy ozone flux. We find strong negative vertical ozone fluxes due to synoptic- and smaller-scale waves in the upper troposphere for both ERA5 and EMAC (Fig. 7b,d), with weaker contributions there due to planetary-scale waves (Fig. 7a,c). These downward fluxes are consistent with down-gradient ozone transport into lower altitudes. In addition, we find weak positive ozone fluxes at the high-latitude tropopause in ERA5, which are primarily due to planetary-scale waves. Furthermore, clear signatures of up-gradient ozone transport due to planetary-scale waves are evident near the STJ core for both ERA5 and EMAC. The physical reasons for this up-gradient ozone transport are further analyzed below. We also note small differences between ERA5 and EMAC in the lower stratosphere (above $\sim$ 350 K). However, the overall contribution of vertical eddy ozone transport to the ozone budget turned out to be negligible in this region (recall Fig. 2).

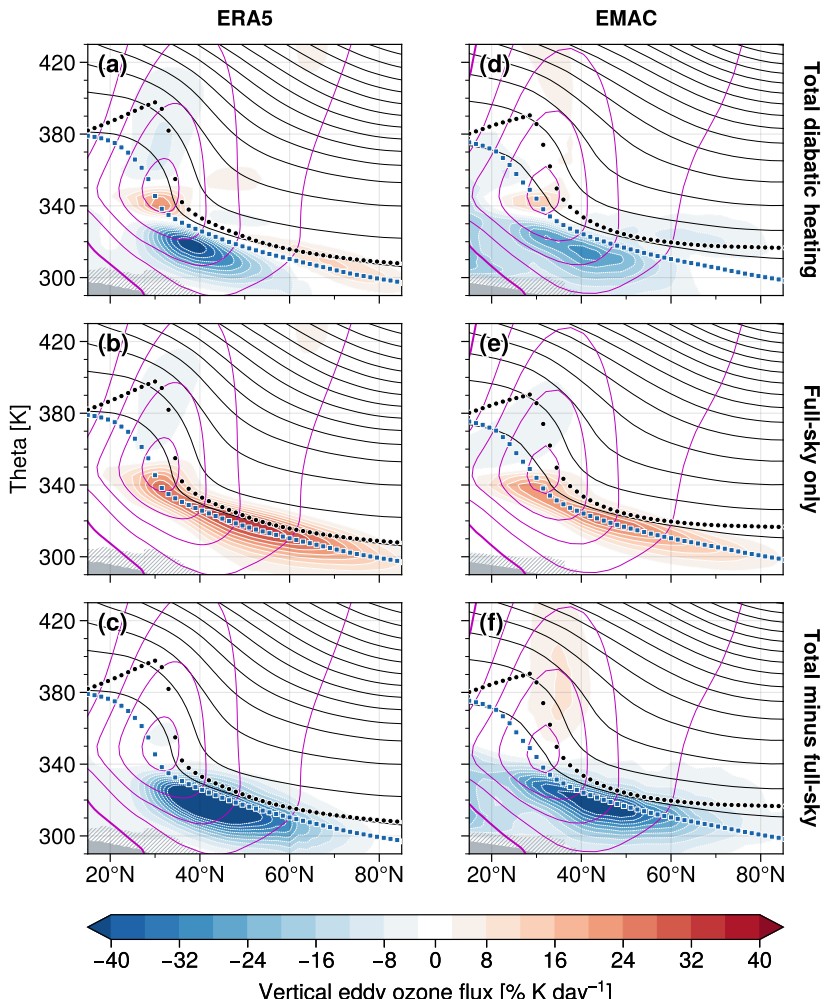

**Figure 8.** Contributions to the vertical eddy ozone flux associated with full-sky radiation and non-radiative effects, respectively, as obtained from (**a**–**c**) ERA5 and (**d**–**f**) EMAC. Other details as in Fig. 7.

The layered transport signatures due to vertical eddy ozone fluxes with alternating ozone tendency maxima and minima in the upper troposphere and tropopause region (cf. Fig. 2) suggest that different diabatic processes may dominate these eddy fluxes in different regions. We therefore decompose the vertical eddy ozone flux into contributions due to different diabatic processes. Figure 8 summarizes these results by showing the winter-mean vertical eddy ozone flux climatologies associated with total diabatic heating, full-sky radiation (long-wave and short-wave) and non-radiative effects (total diabatic heating rates minus full-sky contributions), respectively. It reveals that the complex structure of the total vertical eddy ozone flux arises due to a superposition of simpler structures due to radiative and non-radiative processes. Radiation predominantly causes positive (up-gradient) vertical eddy ozone fluxes, which tend to maximize along the extratropical tropopause (Fig. 8b,e). This is consistent

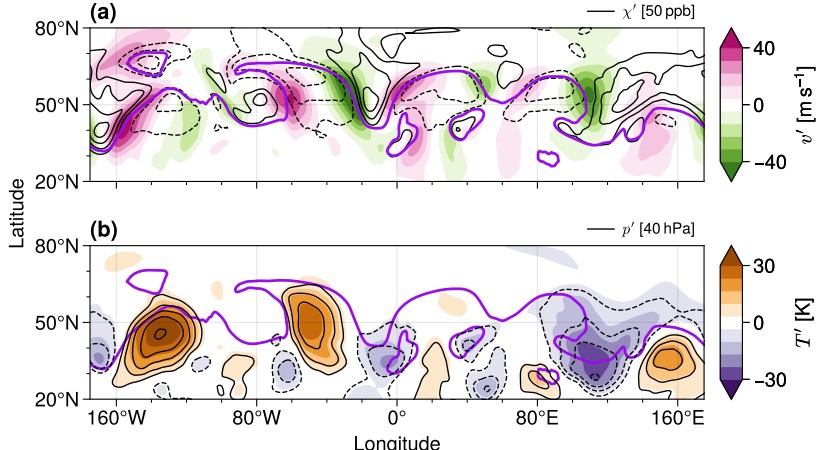

**Figure 9.** Example snapshots for November 28, 2008, to illustrate coupled horizontal and vertical ozone transport due to Rossby waves. (**a**) Zonal deviations of the meridional wind, $v' \equiv v - \overline{v}$ (color coding) and ozone (black contours with 50 ppb contour interval) at 315 K. (**b**) Same as in (**a**) but for temperature (colors) and pressure (contour interval is 40 hPa). The violet thick curve each shows the 2 PVU potential vorticity isoline. Solid (dashed) black contour lines indicate positive (negative) values, where zero contours have been omitted. The data fields have been smoothed by a 5° rolling average along longitude and latitude. ERA5 data at 315 K as obtained from EMCWF's ERA5 catalogue (Hersbach et al., 2017). To provide context for the vertical undulations of the 315 K isentrope, note that air pressure at tropical (equatorward of 30°N) and polar (northward of 60°N) latitudes was approx. 640 hPa and 260 hPa along this isentrope, respectively. Substantial zonal pressure anomalies $p' \sim \mathcal{O}(10^2 \text{ hPa})$ associated with quasi-adiabatic Rossby wave dynamics as shown in (**b**) are therefore reasonable.

for both ERA5 and EMAC, lending support to the robustness of this result. Closer inspection reveals that this is primarily due
to clear-sky radiation associated with planetary-scale waves, i. e., due to processes that we expect to be well-represented in
both data sets. Contributions by cloud radiative effects are small for both ERA5 and EMAC (cf. Fig. S7).

    Other (non-radiative) diabatic processes cause predominantly negative (down-gradient) vertical eddy ozone fluxes, which
tend to maximize in the upper troposphere (Fig. 8c,f). These include latent heating and vertical diffusion, with the former likely
playing a stronger role in the upper troposphere and the latter near and above the tropopause.

Strikingly, the tropopause-level positive vertical eddy ozone fluxes due to radiation would by themselves act to sharpen the
pre-existing strong ozone gradients in this region (by transporting ozone from where it is low, just below the tropopause, to
where it is high, just above the tropopause). Since they are dominated by clear-sky radiation, we consider simple Newtonian
radiative damping as a mechanism giving rise to these positive vertical eddy ozone fluxes as follows (e. g., Andrews et al.,
1987):

$$Q'_{\text{cs}} \simeq -\alpha\theta' = \alpha p' \, \partial_p \overline{\theta} = -\alpha p' \, (g\overline{\rho_\theta})^{-1} \,, \tag{2}$$

where the primed quantities represent the local anomalies from the conventional zonal mean (indicated by overbars). With
that, here the zonal perturbations of clear-sky diabatic heating, $Q'_{\text{cs}}$, are assumed to be proportional to the zonal anomalies

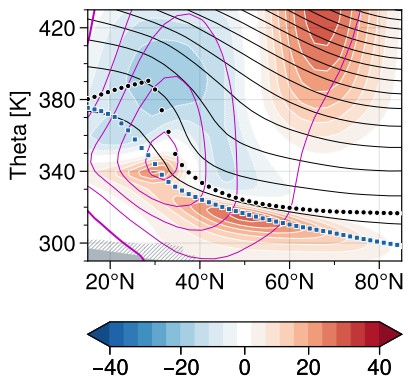

**Figure 10.** Vertical clear-sky eddy ozone flux in EMAC (winter-mean climatology, DJF 2000–2019), $\overline{\hat{Q}_{cs}\rho_\theta\hat{\chi}}/\overline{\rho_\theta}$ in units of $\%\,\mathrm{K\,day^{-1}}$, approximated by the Newtonian damping approach from Eq. (3) with $\alpha = 7$ days. Other details as in Fig. 8.

of potential temperature, $\theta'$, where $\alpha$ denotes the corresponding radiative damping rate. The relation between these potential

temperature anomalies and local changes of air pressure, $p'$, is derived from isentropic density, $\rho_\theta = -g^{-1}\partial_\theta p$ [cf. Eq. (1)].

Equation (2) then shows that upward displaced isentropes ($p' < 0$) result in clear-sky radiative heating, $Q'_{cs} > 0$, and downward

displaced isentropes ($p' > 0$) result in clear-sky radiative cooling, $Q'_{cs} < 0$ (in each case acting against the vertical displace-

ment). Figure 9 illustrates that within Rossby waves, these displacement anomalies are coupled to corresponding ozone anoma-

lies that arise due to horizontal advection as part of the dynamics of the Rossby wave: negative ozone anomalies appear within

high-pressure (anticyclonic) regions and positive anomalies appear within low-pressure (cyclonic) regions. That is, $Q'_{cs}$ and

ozone are positively correlated and the resulting covariance (the vertical eddy ozone flux in isentropic coordinates) is positive

(up-gradient).

For a subsequent analysis with monthly-mean EMAC model output, we approximated the clear-sky vertical eddy ozone

fluxes by combining the budget framework from Eq. (1) and the relaxation approach according to Eq. (2) [notation as before],

$$\overline{\hat{Q}_{cs}\rho_\theta\hat{\chi}} \simeq -\alpha\overline{\hat{\theta}\rho_\theta\hat{\chi}} = -\alpha g^{-1}\overline{\hat{p}\hat{\chi}}^*, \tag{3}$$

which is shown in Fig. 10. The close agreement in the structures and magnitudes of this approximation and the actual clear-sky

vertical eddy fluxes (cf. Fig. 8e) along the tropopause provides evidence that long-wave radiative damping indeed can explain

most parts of the up-gradient ozone flux signatures along the tropopause under clear-sky conditions with a typical relaxation

rate $\alpha^{-1} \sim 1$ week. The weak positive ozone fluxes associated with cloud radiative effects at the tropopause (cf. Fig. S7) can

likely be explained in a similar way by considering cloud-top long-wave cooling that is linked to Rossby wave dynamics in the

mid-latitudes. We furthermore investigated a passive, linear Age of Air tracer in EMAC, where we found similar up-gradient

vertical eddy fluxes along the tropopause (not shown), suggesting that short-wave ozone–radiation feedback plays a rather

minor role in causing this feature.

Finally we note that for Boreal summer (June–August), the down-gradient vertical eddy ozone fluxes in the upper troposphere

associated with non-radiative effects turned out to be substantially stronger than during winter (cf. Fig. S6 in the supplemental

material). In contrast, the up-gradient vertical eddy fluxes under full-sky conditions in the tropopause region are much weaker, likely due to the reduced vertical gradients of ozone and potential temperature across the tropopause. The total contribution of vertical eddy transport to the UTLS ozone budget during summer is therefore mostly governed by down-gradient ozone transport (cf. fourth row in Fig. 4).

## 5    Summary and conclusions

In this paper, we studied different aspects of the zonal-mean ozone budget of the northern hemispheric lowermost stratosphere (LMS). Our comparison of winter-mean ozone transport climatologies based on 20 years of ERA5 reanalysis data and EMAC climate model output confirmed the expected important contributions by quasi-horizontal adiabatic eddy mixing and diabatic downward advection by the residual stratospheric circulation. These two contributions together well reproduce the total winter-mean ozone tendencies in the upper extratropical LMS. However, near the tropopause and the subtropical jet core the effects

of horizontal mean flow advection and vertical eddy ozone transport become relevant. We noted differences in the actual magnitudes of the individual transport contributions and ozone background gradients between ERA5 and EMAC, which may be associated with the model's intrinsic resolution, parameterization schemes and approximations that govern each model's dynamical transport characteristics, as well as numerical diffusion that may have been partly corrected for by data assimilation only in the reanalyses.

Our analysis furthermore showed that meridional eddy ozone transport in the upper LMS (i. e., the shallow branch of the Brewer–Dobson circulation, BDC) is mainly governed by medium- to smaller-scale waves and is strong throughout the whole year due to continuous dynamical wave driving. This is in contrast with poleward ozone transport in the higher stratosphere (as part of the deep branch of the BDC) that is mostly driven by planetary-scale waves and shows strong seasonality with strongly reduced horizontal eddy transport during summer. It is worth noting that wave-driven meridional mean flow advection

contributes significantly in the subtropical lower stratosphere (across the upper flank of the subtropical jet) and therefore seems to be clearly part of the shallow branch of the BDC.

    The winter-mean climatology of the horizontal eddy ozone flux supports the concept of diffusive, down-gradient eddy transport in the UTLS, acting to reduce the underlying ozone background gradients. This is not the case, however, for vertical eddy ozone transport, especially near the tropopause, where the corresponding ozone flux climatologies clearly indicate upward

(up-gradient) transport, which by itself would act to sharpen the pre-existing positive vertical ozone gradients at the tropopause. We found these upward eddy ozone fluxes to arise primarily due to radiative damping within planetary-scale Rossby waves. Since these Rossby waves are at the same time responsible for the down-gradient horizontal eddy fluxes, which are stronger in magnitude, these radiatively-damped vertical fluxes should be viewed as effectively reducing the horizontal fluxes.

    Significant vertical eddy ozone fluxes are also found due to latent heating in the upper troposphere, and to a lesser degree

due to vertical diffusion near the tropopause, leading to down-gradient vertical ozone transport. Cloud-radiative effects, which usually come along with substantial uncertainties in general circulation models, and ozone–radiation feedback turned out to be less important in the analyzed models. For northern hemispheric summer, we found much weaker signatures of up-gradient

vertical ozone fluxes, which is probably in part due to the weaker vertical gradients of ozone across the tropopause during that time. Consequently, vertical eddy ozone transport during summer is dominated by down-gradient transport in the upper troposphere through synoptic- and small-scale waves. In general, these findings suggest that our picture of eddy ozone transport in the tropopause region should include both isentropic, quasi-horizontal eddy mixing as well as the coupled diabatic effects associated with Rossby wave dynamics that can contribute substantially to the local ozone budget.

*Code and data availability.* ERA5 reanalysis data on model levels can be downloaded at https://apps.ecmwf.int/data-catalogues/era5/?class= ea. EMAC model output and code for processing the data used in this study are available from the authors upon request. This paper contains modified Copernicus Climate Change Service information 2025. Neither the European Commission nor ECMWF is responsible for any use that may be made of the Copernicus information or data it contains.

*Author contributions.* FH conducted the analyses and prepared the first draft of the manuscript. He was supervised by HG and TB. FP provided pre-processed ERA5 reanalysis data on model levels and advice for the subsequent data processing. MM set up and ran the EMAC simulation and provided the model output. All authors helped to improve the manuscript.

*Competing interests.* The authors declare that they have no conflict of interest.

*Acknowledgements.* We thank Kris Wargan and one anonymous reviewer for their valuable comments and suggestions, which significantly helped to improve the manuscript. We thank P. Conrat Fuentes (LMU Munich) for his work on Python code for computing thermal tropopause heights, which was partially used within this study. We furthermore thank P. Jöckel (DLR) for his support in updating and implementing the EMAC model setup. We acknowledge the usage of the Python programming language and several extensions included therein, in particular, among others, the software packages xarray (Hoyer and Hamman, 2017) and PyTropD (Adam et al., 2018; Ming, 2022). The figures in this paper have been produced using Matplotlib (Hunter, 2007) and ProPlot (Davis, 2021). This work was funded by the Deutsche Forschungsgemeinschaft (DFG, German Research Foundation) – TRR 301 – Project-ID 428312742. We acknowledge the computing time on the JUWELS supercomputer at the Jülich Supercomputing Centre (JSC), which was granted under the VSR Project-ID "CLAMS-ESM", and technical support by the JSC staff. This work used resources of the Deutsches Klimarechenzentrum (DKRZ) granted by its Scientific Steering Committee (WLA) under Project-ID "bd1305".

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
