# Peer review of "Adiabatic versus diabatic transport contributions to the ozone budget in the northern hemispheric upper troposphere and lower stratosphere"

_EGUsphere, 2025_

## Author Comment (AC1)

**Author response**

**Adiabatic versus diabatic transport contributions to the ozone budget in the northern hemispheric upper troposphere and lower stratosphere**

Frederik Harzer, Hella Garny, Felix Ploeger,
J. Moritz Menken and Thomas Birner

September 3, 2025
* * *
We thank both reviewers for their detailed feedback and for their valuable suggestions that greatly helped to improve the presentation of our results in the manuscript. In the following, the reviewers' comments are printed with blue color and are marked by indentation. Our responses each are inserted directly beneath the corresponding remarks, where we refer to the line numbering of the tracked-changes document.

**Reviewer #1**

> This paper uses ERA-5 and EMAC output to provide a detailed analysis of the contributions to the lowermost stratosphere (LMS) zonal mean ozone transport from advection and eddy mixing. In addition to the known dominant role of diabatic advection and isentropic mixing, the authors demonstrate that horizontal advection and vertical mixing play a non-negligible role near the tropopause. Furthermore, they separate the vertical eddy fluxes into radiative and non-radiative components and identify signatures of up-gradient mixing near the subtropical jet stream. I believe those are novel results that will be of interest to ACP readers. The paper is very well written. The findings are clearly supported by figures, including those in the supplementary material. I have only a few very minor comments and suggestions.

Thank you for your comprehensive feedback!

> **Minor comments**
>
> Please, define all symbols used in equations 1–3.

Thank you, we agree that more details on the symbols and notation used clearly provide more guidance to the reader and improve readability. We now added more explanations for the first two equations (ll. 104–113, 269–273) and added a note that Eq. (3) directly follows from these and thus uses the same notation as before (ll. 280–282).

> L34. "Quasi-observed" is an interesting term that I don't think I've seen before. I take it to mean that reanalysis circulation, while not directly observed, is tightly constrained by data. I'm not sure if it will be clear to readers less familiar with data assimilation. I like the term, but perhaps one sentence of explanation would be useful.

Thank you, we inserted a footnote with more details on the interpretation of this term (l. 36).

> L56. MLS is assimilated starting 2004 but this analysis uses the 2000–2019 period. Are the results from 2000–2004 as reliable as in the later period? I would think that assimilation of MLS makes a big difference especially in the UTLS.

We agree that the lack of MLS data prior to 2004 is expected to cause more uncertainties in the reanalysis data. We added a corresponding note in the main text now (ll. 59–61). Furthermore, we briefly compared the ozone tendencies in Fig. 1 prior to and after 2004, respectively, but did not find major structural differences between these two time periods (cf. Fig. R1 at the end of this author response). Finally, the overall good agreement with the independent EMAC model output supports the reliability of ERA5.

> L58. How long are these forecasts? Presumably very short (several hours?).

Thank you, we revised the corresponding paragraph in the manuscript to make it more precise (ll. 62–67).

> LL58–62. More about this. Aren't the temperature tendencies constrained by assimilation to some extent? Long wave cooling depends strongly on temperature, which is assimilated.

We definitely agree that this detail should be taken into account. We slightly adapted the corresponding expression, now reading "are not subject to direct assimilation of observational data" (l. 63), but decided to not provide more elaboration to avoid distraction from the main point.

> L67-68. Does it mean that the spinup was 2000–2001 or 1998–1999?

Thank you for your comment. We agree that this sentence needs some more clarification: the simulation was initialized two years before the investigated time period. The sentence has been revised accordingly (l. 74).

> Figure 1 caption. Please, specify which dotted line marks which tropopause. I'm guessing the blue dotted line is the 2-PVU one.

Thank you, we revised the figure's caption accordingly.

> LL96–100. All symbols used in Equation 1 should be defined in the text.

Done. Thank you!

> Figure 3. Would it make sense to include analysis tendencies in panels a and b? Would the sum of all the terms (and the analysis tendency) reproduce the "total ozone tendency" shown in gray? It looks to me like in EMAC total transport (and horizontal eddy transport) is balanced by chemistry near the STJ. I wonder if analysis tendencies play an analogous role in ERA-5 to produce a "total tendency" profile very similar to that in EMAC. I also wonder if the black line in Fig. 3a would look more like 3c if the analysis tendency was included in it. The differences between panels a and b between 20N and 40N are substantial (even the sign of the net transport tendency is different) and require more explanation than what's provided in the text.

We definitely agree that this is a very interesting point that would be worth a closer look. Unfortunately, we cannot conduct this analysis at this point in time without considerable additional efforts, since so far we have not processed the corresponding analysis increments for ERA5. Concerning the differences between ERA5 and EMAC at (sub-)tropical latitudes, we hypothesize that these may be due to model differences in the large-scale tropical circulation and the tropical tropopause layer (TTL). We only added a short note on this to the manuscript (ll. 165–171), since the subtropical region around 20°N to 40°N is not the main focus of our study.

> Figure 3. The gray shading should be described in the caption, not just in the main text. Also, is it possible to call this something other than "total ozone tendency", which suggests vertically integrated ozone?

We now refer to this as the actual ("observed") tendency derived from the zonal-mean ozone distribution and added more details to improve the clarity of this figure and the corresponding paragraph in the main text (ll. 152–155).

> L213. This is very interesting. Up-gradient eddy transport seemingly contradicts the statement made in lines 201–202 about eddy transport (horizontal, in that case) being generally down-gradient. Perhaps it would be worth mentioning right here that this is explained later in the discussion of Figure 9, because eddy transport is diffusion-like and uphill diffusion is counterintuitive.

We see your point and added a short note in the manuscript (l. 245).

> Equation 2. Again, most symbols are undefined.

Done. Thank you!

> LL244-248. I think this comparison should be explicitly shown in a figure to make the argument more quantitative.

We added the results of this analysis based on EMAC model output as new Figure 10.

**Technical corrections**

L31. "our knowledge on..." □ "our knowledge of..."

L252: "rather plays a minor role" □ "plays a rather minor role"

We revised the manuscript accordingly (ll. 32 and 291). Thank you!

**Reviewer #2**

This is a nice and interesting paper that improves our understanding of the contributions of different transport processes (in particular advection and mixing) to the total ozone budget in the extratropical UTLS region. The authors show very clearly how vertical advection and horizontal eddy transport are the most important contributors, although they demonstrate that horizontal advection and vertical mixing also play an important role along the extratropical tropopause. In addition, they analyse radiative and non-radiative effects on the vertical eddy ozone flux, finding that the upward eddy ozone fluxes near the tropopause and the STJ core are mainly due to radiative dumpling within planetary-scale Rossby waves.

Overall, I consider the paper to be very well-written, with several very nice and novel results well supported by the figures. I consider it to be of interest to ACP readers and I recommend it for publication after addressing a few minor comments and suggestions which I list below.

Thank you for your detailed review!

- Line 24: Perhaps it would be a good idea to include here the recent paper by Benito-Barca et al. 2025 (https://doi.org/10.1029/2024JD042412), which confirms the importance of transport processes in ozone trends in the most recent CCMs (CCMI-2).

We agree that this publication complements our discussion on ozone trends and added this reference at the end of the corresponding paragraph (l. 24). Thank you!

- Line 67: This is a bit confusing... the spin-up years are 1998-99 or 2000-2001? Please clarify.

Thank you for your comment. We agree that this sentence needs some more clarification: the simulation was initialized two years before the investigated time period. The sentence has been revised accordingly (l. 74).

- Line 68: Just out of curiosity, why do you use ERA5 to prescribe SSTs and SICs instead of observations (e.g. HadISST or ERSST)?

Thank you for your question. We agree that SSTs and SICs could be used directly from observations. However, this was not done in this case since a "twin" simulation was run together

with the simulation considered in this study, which is intended for use in other ongoing work. Both simulations feature identical setups except that the twin simulation applies Newtonian relaxation ("nudging") towards ERA5 reanalysis data, such that also SSTs and SICs are used from ERA5 for consistency. To isolate the effects of nudging, the same SST/SIC forcing is used in both simulations. The pair of these two simulations is currently used in a study by J. Moritz Menken to investigate the impact of nudging on the simulation results on climate scales. Although nudging is important to allow for direct comparison between observations and model data, its effects on long-term trend projections and the model's variability is not yet clear. The preparation of the corresponding manuscript is currently ongoing.

> ■ Line 85: There is something here that I have not fully understood. In line 48 it says that ERA5 has been interpolated to match the resolution of the model. However, now here it says that the model has been interpolated to the same levels as ERA5, which "led to slight vertical oversampling of the EMAC data". Why isn't ERA5 interpolated directly to the same levels as the model, instead of interpolating the model to ERA5 levels? I wonder if the vertical oversampling of the model could cause problems in the region near the tropopause. I think this point needs to be better clarified, as understanding the vertical resolution of each dataset is important for interpreting the results in the UTLS region.

We initially started to interpolate ERA5 data on model levels with full vertical resolution since we intend to use this dataset for various research projects. We later found that our analysis of LMS ozone with ERA5 may benefit from adding an independent EMAC simulation for comparison. Although we interpolated the EMAC model output on the same isentropic levels as for ERA5, the vertical distribution of the EMAC model levels across the extratropical LMS is still very dense, such that we did not expect major differences due to this approach. Indeed, from a brief comparison of the ozone tendencies presented in Fig. 1, computed with a vertical resolution similar to either ERA5 or EMAC, we obtained almost equivalent results (cf. Fig. R2 below). We added a corresponding note to the revised manuscript (l. 95).

> ■ Caption Figure 1: The thermal tropopause is indicated by the blue dotted line and the 2 PVU isolines by the black one, right? Please indicate in this caption.

Done. Thank you!

> ■ Line 99: It is not clear to me what you mean by 'overbars (with asterisks)'. What overbars without asterisks indicate? Also, there are symbols that are not defined. Please define all symbols in the equation.

We revised the description of this equation and added more details on the symbols and notation used (ll. 104–113). Thank you!

■ Figures 2b and 2f: I am curious about the negative ozone tendency on the equatorial flank of the subtropical jet in EMAC, which does not appear in ERA5. Do you have any idea why this might be happening?

The strong negative ozone tendency in EMAC (Fig. 2f) near the core of the subtropical jet stream coincides with the increased equatorward advection through the meridional mean flow in Fig. 2g compared to ERA5 (Fig. 2c). We therefore think that these may be caused by differences in the tropical tropospheric circulation and the representation of tropospheric ozone between ERA5 and EMAC. From a preliminary analysis provided in Fig. R3 we indeed found evidence for stronger wave propagation from the equator towards subtropical latitudes in the upper troposphere in EMAC compared to ERA5, which may hint at increased equatorial Rossby wave activity. We added a brief discussion to the manuscript in ll. 142–148.

■ Line 124: please define STJ (here is the first mention).

Done (l. 137). Thank you!

■ Line 137: Here, in addition to the differences between EMAC and ERA5, I think it is important to mention that the challenge of this region can also be seen in the difference between the ozone tendency obtained directly from the actual ozone distribution and the one obtained through the sum of the transport and chemical terms. I get the impression that equation (1) closes the total ozone budget reasonably well north of 40°N but has problems in the subtropics.

We revised the corresponding paragraph and extended the discussion of this figure (ll. 142–171). Please also consider our response on a similar comment by Reviewer #1 on p. 3 of this document.

■ Lines 176-178: my impression is that for large-scale waves (WV 1-3) there is also a stronger transport during winter and spring.

We revised this sentence to make it more precise (ll. 206–210). Thank you!

■ Line 236/ Equation 2: For those who are unfamiliar with these equations, it would be convenient to define all the symbols.

Done (ll. 269–273). Thank you!

■ Caption of Figure 9: Is the black contour interval in Figure 9b in hPa or Pa units? Is it possible that there are anomalies of more than 100 hPa? It sounds strange to me...

Thank you for this comment. Figure 9 displays zonal anomalies along the 315 K isentrope, which features strong meridional gradients in geometric altitude and pressure. That is, air pressure at 315 K varies between approx. 640 hPa at tropical latitudes (equatorward of 30°N)

and approx. 260 hPa in the polar regions (poleward of 60°N). This allows for pressure anomalies of more than 100 hPa due to quasi-adiabatic Rossby wave dynamics. We added a corresponding note on this in the caption of Fig. 9.

> ■ Line 248 (and also 285): I think this result is interesting enough to include a figure to see it explicitly (at least as supplementary material).

Thank you, we added a new Fig. 10 showing the approximation of the EMAC clear-sky vertical eddy fluxes by the radiative damping approach from Eq. (3). Since the contribution by cloud radiation effects is negligibly small both in ERA5 and EMAC, we added this analysis as Fig. S7 to the supplement for the interested reader.

[Figure]

**Figure R1:** Same as Fig. 1 in the manuscript but for ERA5 only and for two time periods covering monthly-mean data from 2000 through February 2004 (left column; includes 4 DJF seasons) as well as December 2004 until 2019 (right column; includes 15 DJF seasons), respectively.

[Figure]

**Figure R2:** Same as Fig. 1 in the manuscript but now computed from ERA5 and EMAC data on potential temperature levels with somewhat reduced vertical resolution similar to that of the intrinsic EMAC model levels in the LMS.

[Figure]

**Figure R3:** Winter-mean ERA5 and EMAC climatologies (DJF, 2000–2019) of the meridional mass streamfunction (black contours) in units of $10^9$ kg s$^{-1}$ (calculated using PyTropD by Adam et al., 2018; Ming, 2022), Eliassen–Palm (EP) flux (shown by the arrows) and EP flux divergence (color coding). The thick black dotted lines show the winter-mean thermal tropopause (computation as done in the manuscript). EP formalism and plotting as in Andrews et al. (1983) and Jucker (2021), respectively.

**References**

Adam, O., Grise, K. M., Staten, P., Simpson, I. R., Davis, S. M., Davis, N. A., Waugh, D. W., Birner, T., and Ming, A.: The TropD software package (v1): standardized methods for calculating tropical-width diagnostics, Geosci. Model Dev., 11, 4339–4357, https://doi.org/10.5194/gmd-11-4339-2018, 2018.

Andrews, D. G., Mahlman, J. D., and Sinclair, R. W.: Eliassen-Palm Diagnostics of Wave–Mean Flow Interaction in the GFDL "SKYHI" General Circulation Model, J. Atmos. Sci., 40, 2768–2784, https://doi.org/10.1175/1520-0469(1983)040⟨2768:ETWATM⟩2.0.CO;2, 1983.

Jucker, M.: Scaling of Eliassen-Palm flux vectors, Atmos. Sci. Lett., 22, https://doi.org/10.1002/asl.1020, 2021.

Ming, A.: PyTropD version 2.12, GitHub, https://github.com/TropD/pytropd, 2022.